# Bacterial diversity and antibiotic susceptibility profile of the isolates from public vehicles in Butwal Sub-Metropolitan City, Rupandehi, Nepal

Ram Bahadur Khadka *, Balram Neupane, Khimdhoj Karki, Saksham Pokharel

Department of Medical Laboratory Science, Crimson College of Technology (Affiliated to Pokhara University), Butwal, Nepal

* rambahadurkhadka00@gmail.com

## Abstract

Public transportation systems are vital for urban mobility but are susceptible to bacterial contamination due to high passenger density, frequent shared surface contact,minimal ventilation, and unsatisfactory cleaning practices promoting transmission of pathogenic and antibiotic resistant bacteria.,The main aim of this study was to explore the bacterial diversity associated with selected public vehicles and determine the antibiotics susceptibility patterns of bacterial isolates.A descriptive cross-sectional study was carried out from July 2023 to July 2024 at Paschimanchal Bus Terminal Stop, Butwal Sub-Metropolitan City, Rupandehi, Nepal. A total of 324 swabs were gathered from high-contact surfaces in AC buses, micro buses, and wingers using systematic sampling. Standard laboratory methods were used to identify bacterial cultures and their antibiotic susceptibility.Out of 324 surface swabs from 81 public vehicles, 262 (80.86%) showed bacterial growth. Among 262 isolates 148 (56.48%) were Gram positive and 114 (43.51%) Gram-negative. Predominant Gram positive isolates were *CoNS* 72 (27.48%), *S.aureus* 52 (19.84%) and *Bacillus spp.* 21 (8.02%), while Gram negative included *E.coli* 32 (12.21%), *Klebsiella spp.* 26 (9.92%) and *Pseudomonas spp.* 21 (8.02%). *CoNS* (44.44%, 32/72) and *S.aureus* (42.31%, 22/52) were significantly more frequent in Micro buses (P < 0.05). High antibiotic sensitivity was found for Gentamicin (100%, 12/12 for *Micrococcus* sp.; 97.22%, 70/72 for *CoNS*) and Meropenem (100%, 26/26 for *Klebsiella spp.)*. Resistance was highest to Cetriaxone (100%, 9/9 for *Citrobacter spp*.) and Amoxicillin-clavulanic acid (69.2%. 18/26 for *Klebsiella spp*.) The extensive bacterial contamination present indicates an immediate need for better hygiene practices, improved cleaning methods, and public education to limit the microbial hazards associated with public transportation. Reduced sensitivity to commonly prescribed antibiotics, such as erythromycin in *Staphylococcus aureus*, indicates emerging antimicrobial resistance, posing greater risks to public health through limited treatment options and the spread of resistant strainsin the community.

**Data availability statement:** All data underlying the findings of this study are fully available without restriction. The complete dataset, including all raw data points used in the analysis, has been deposited in the Zenodo public repository. https://doi.org/10.5281/zenodo.17766523.

**Funding:** This work was supported by the Pokhara University Research Center (PURC) (Approval/Grant No. 02-2079/80) awarded to Ram Bahadur Khadka. The funders had no role in study design, data collection and analysis, decision to publish, or preparation of the manuscript. No commercial companies provided funding for this study, and RBK received salary support from Crimson College of Technology during the study period. The funder's website is: https://pu.edu.np/workplaces/purc/.

**Competing interests:** The authors have declared that no competing interests exist.

## Introduction

Public transportation systems facilitate mobility and connectivity in urban environments worldwide [1]. However, alongside their convenience, these shared spaces also harbor a hidden threat of bacterial contamination. From buses to taxis, tricycles to metro-buses, the surfaces of public vehicles serve as fertile breeding grounds for a diverse array of microorganisms, including potentially harmful pathogens [2]. Studies conducted across different regions including Istanbul city (Turkey) and Quito city (Ecuador) have consistently highlighted the presence of elevated levels of microbial diversity within these vehicles, raising concerns about the implications for public health [3,4].

The findings of these studies underscore the pervasive nature of bacterial contamination within public transportation systems. For instance, research conducted in Istanbul, Turkey, revealed that handles and surfaces in buses and metro buses were teeming with *Staphylococcus aureus, coagulase-negative staphylococcus, and Enterococcus spp.,* highlighting the potential for these vehicles to serve as reservoirs for harmful microbes [3,5]. Similarly, studies conducted in other regions, such as Ghana and Nepal, have identified the presence of *Escherichia coli, Salmonella spp., and methicillin-resistant Staphylococcus aureus (MRSA)* on surfaces within public transport vehicles, further emphasizing the widespread nature of bacterial contamination [6–8].The implications of bacterial contamination in public transportation extend beyond immediate health risks to passengers. The presence of pathogenic bacteria raises concerns about the potential for disease transmission within communities, particularly in densely populated urban areas [9].

Public transport plays a significant role in the spread of antibiotic-resistant bacteria in cities due to the high number of people using these systems daily [10]. Understanding how these bacteria respond to antibiotics is crucial for planning effective strategies to prevent infections. A prior investigation reported number of drug resistance bacteria including *Staphylococcus aureus, Klebsiella pneumoniae, Escherichia coli* and *Enterobacter spp.* on public vehicles in Dhaka City, Bangladesh [11]. Marked similarly, studies in Mekelle, Ethiopia [12], and in Portland, USA [13] identified antibiotic resistant bacterial isolates from public transport systems. These studies further emphasize the worldwide nature of the public health problem. *Staphylococcus aureus*, including methicillin-resistant *S. aureus* (MRSA), has been found in buses and subways, with about 65% of MRSA strains showing resistance to multiple antibiotics [14]. Similarly, *Klebsiella pneumoniae* resistant to carbapenems and *E. coli* resistant to colistin due to the mcr-1 gene have been detected. Other bacteria, such as *Enterococcus faecium* and *Enterococcus faecalis*, are also common on public transport surfaces and often show resistance to several antibiotics [15–17].

Addressing the issue of bacterial contamination in public transportation requires a comprehensive approach. This includes the implementation of rigorous cleaning and disinfection protocols, as well as the promotion of personal hygiene practices among passengers. Additionally, the development of innovative solutions, such as antimicrobial surface coatings, holds promise for reducing the prevalence of pathogenic bacteria in these environments [18,19]. Furthermore, raising awareness among the

general population about the risks associated with microbial contamination in public transport is essential in promoting behavior change and fostering a culture of cleanliness and hygiene [13,20]. Therefore the present study aimed to determine Bacterial Diversity and Antibiotic Susceptibility Profile of the Isolates from Public Vehicles in Butwal Sub-Metropolitan City, Rupandehi, Nepal.

## Materials and methods

### Ethics statement

This present study was conducted on environmental samples, collected from public vehicles. No human participants or animals were involved in the research. The individuals pictured in the supporting information files have provided written informed consent (as outlined in PLOS consent form) to publish their image.Research approval for this study was obtained from the Pokhara University Research Center (PURC),Pokhara University,Kaski,Nepal (Approval/Grant No. 02-2079/80).

### Study site

A descriptive cross-sectional study was conducted at Paschimanchal Bus Terminal Stop, Butwal Sub-Metropolitan City, Rupandehi, Nepal, from July 2023 toJuly 2024. This urban transportation hub serves as a central point for commuters traveling within the city, neighbouring cities, and districts.

### Collection and processing of the sample

A systematic sampling approach was utilized to collect swab samples from surfaces within the selected public vehicles at the Paschimanchal Bus Terminal Stop in Butwal Sub-Metropolitan City, Rupandehi, Nepal. A total of 324 swabs were collected from various high-contact surfaces, including steering wheels (S), handles (H), seat backs (SB), and seat handles (SH), across different types of vehicles such as AC Buses, Micro Buses, and Wingers. Swab specimens were collected using sterile cotton-tipped applicator sticks moistened with sterile normal saline. These swab samples were aseptically inserted into separate sterile test tubes, labeled, and transported to the Microbiology Laboratory of Crimson College of Technology, Butwal, using an icebox within 1 hour for immediate processing [21–24].

### Identification of bacteria

Bacterial culture and identification were carried out by inoculating swabs collected from public vehicles surfaces onto various culture media (HiMedia Laboratories Pvt.Ltd.) like Nutrient Agar (NA), Mannitol Salt Agar (MSA), McConkey Agar (MAC), Blood Agar, Thiosulfate-Citrate-Bile Salts-Sucrose (TCBS), and Xylose Lysine Deoxycholate (XLD) agar [21,22]. Culture plates were incubation at 37 $^0$C for 24–48 hours, depending on the bacterial species. After incubation morphological examination of bacterial colonies were conducted targeting colonies size, shape, color, margin, elevation and haemolytic patterns. Gram staining was conducted to differentiate gram-positive and gram-negative bacteria. Bacterial colonies were enumerated, and further confirmation of bacterial species was carried out through specific biochemical tests.Triple Sugar Iron (TSI) test was used to assess sugar fermentation and hydrogen sulphide producton, Voges-Proskauer (VP) and Methyl Red (MR) tests are used to detect metabolic end production; Indole test evaluated tryptophan degradation; Catalase,Oxidase and Citrate tests assessed bacterial specific enzymatic activities. All carried out tests were performed by following the microbiological procedures outlined in Bergey's Manual of Bacteriology [21].

### Antibiotic susceptibility test (AST) of the bacterial isolates

Antibiotic susceptibility testing (AST) was performed using the modified Kirby-Bauer disk diffusion method, adhering to the guidelines of the Clinical and Laboratory Standards Institute (CLSI) [25]. Antibiotic discs (HiMedia Laboratories Pvt.Ltd.)

were chosen based on the type of bacterial isolates, with Gram-negative bacteria tested using discs containing Amikacin (30 µg), Cefoxitin (30 µg), Chloramphenicol (30 µg), Ciprofloxacin (5 µg), Imipenem (10 µg), Piperacillin/tazobactam (TZP) (100/10 µg), Erythromycin (15 µg), Gentamicin (10 µg), Ceftriaxone (30 µg), and Co-trimoxazole (25 µg). For Gram-positive bacteria, the discs used were Amikacin (30 µg), Amoxicillin-clavulanic acid (20/10 µg), Cefotaxime (30 µg), Ceftazidime (30 µg), Ceftriaxone (30 µg), Chloramphenicol (30 µg), Ciprofloxacin (5 µg), Co-trimoxazole (25 µg), Meropenem (10 µg), and Gentamicin (10 µg). Bacterial isolates were inoculated onto Mueller-Hingon agar culture plates and antibiotic discs were applied. Mueller-Hinton agar plates were incubated at 35±2⁰C for 16–18 hours.The diameter of the zones of inhibition around each disc were measured in millimetres and susceptibility was interpreted according to the CLSI 2022 breakpoints guidelines to classify the isolates as *susceptible (s), intermediate (I) or resistant (R)* for each antibiotic and bacterial species [21,26,27].

### Quality assurance

Aseptic techniques were used in all the steps of specimen collection and inoculation to minimize contamination. Specimens were collected in aseptic conditions. Reagents and antimicrobial discs were checked for expiry date. Sterility of culture media was carried out by incubating 5% of the prepared media before inoculation. *Escherichia coli (ATCC 25934), Staphylococcus aureus (ATCC 25955),* and *Pseudomonas aeruginosa (ATCC 27866)* reference strains were used to control the performance of culture media and antibiotic discs to assess the quality of the general laboratory procedure.

### Data analysis

Data were entered into Microsoft Excel and imported into Statistical Package for Social Sciences (SPSS) software version 21.0 for analysis. Descriptive statistics, including frequencies, percentages and proportions were calculated to summarize variables like as sample sources, vehicle types, bacterial species and antibiotic sensitivity patterns. Means and standard deviations applied to summarize for continuous variables.The chi-square test was used to determine the associations between categorical variables to identify whether bacterial contamination differed by public vehicle type and weather antibiotic resistance patterns varied among isolated bacterial species.A *p-value* ≤0.05 was considered as a statistically significant exploring a meaningful association between the tested variables (S1 Data).

## Results

Over a one year period, a comprehensive investigation was conducted involving 81 public vehicles, comprising AC (Air Conditioner) Buses (n=27), Micro Buses (n=27), and Wingers (n=27). A total of 324 swabs were meticulously collected from high contact surfaces specifically steering wheels (S), handles (H), seat backs (SB), and seat handles (SH).These locations were chosen because they are the surfaces most frequently touched by passengers and drivers, thereby presenting the highest risk of bacterial contamination due to frequent contact. Out of the 324 swabs collected, bacterial growth was observed in 262 specimens, accounting for a prevalence rate of 80.86%.

### Frequency of bacterial isolates

Among the 262 bacterial isolates obtained from the swab samples, Gram-positive bacteria accounted for 148 (56.48%) of the total isolates, while Gram-negative bacteria constituted 114 (43.51%) of the isolates. The most prevalent Gram-positive bacteria species included *Coagulase-negative staphylococci (CoNS)* 72(27.48%), *Staphylococcus aureus* 52(19.84%), *Bacillus spp.*13 (8.02%), and *Micrococcus spp.*11(4.19%). Notable Gram-negative bacteria isolate comprised *Escherichia coli* 32 (12.21%), *Klebsiella spp.*26 (9.92%), *Pseudomonas spp.*13 (8.02%), *Proteus spp.*13 (8.02%), *Enterobacter spp.*11 (4.19%), *Acinetobacter spp.*10 (3.81%), and *Citrobacter spp.* 9 (3.43%) (Fig 1).

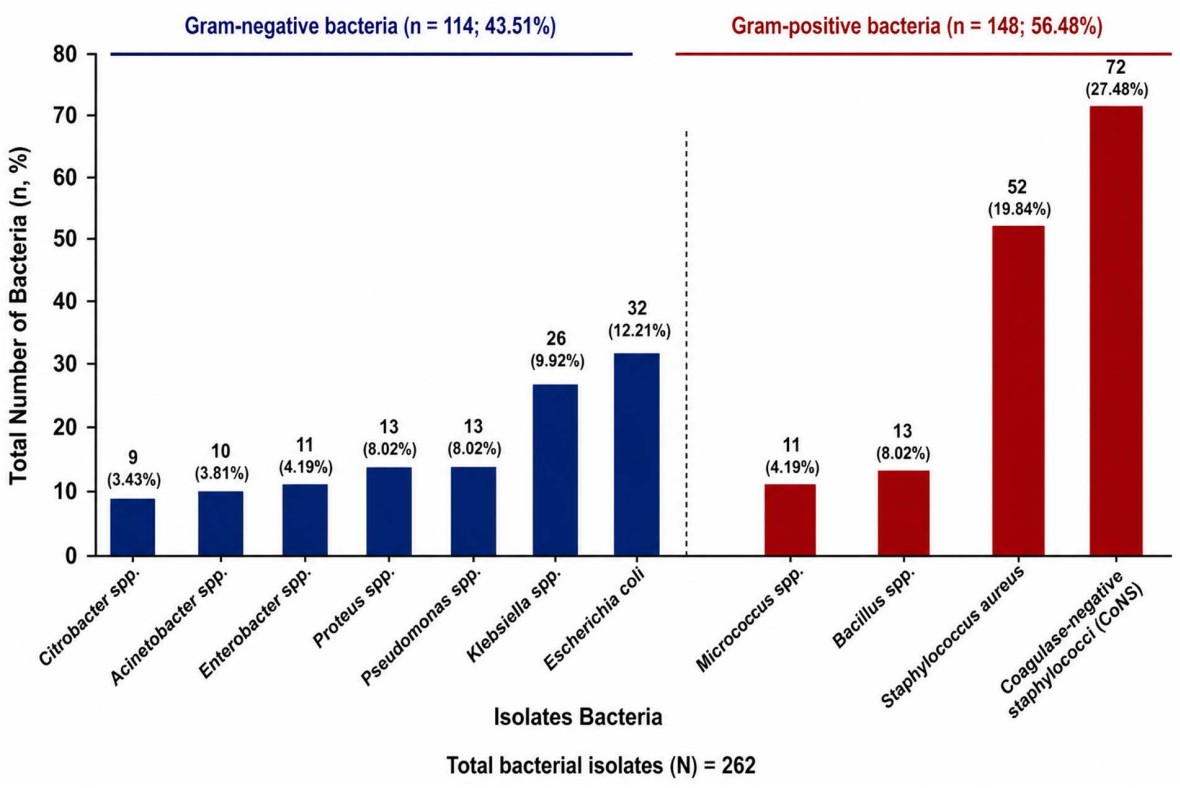

**Fig 1. Frequency of Bacterial Isolates – This figure illustrates the distribution of various bacterial isolates obtained from swab samples.**

## Distribution of bacterial isolates between AC bus, micro bus, and winger

The distribution of bacterial isolates across different types of public vehicles, including AC Buses, Micro Buses, and Wingers, varied significantly as indicated by the calculated P-values. *Coagulase-negative Staphylococci (CoNS)* were presentig significantly greater degrees in the Micro Buses 32 (44.44%) than AC Buses 14 (19.44%) and Wingers 26 (36.11%) (P<0.001). The findings can be better understood through evaluation of the proportion of types of *coagulase negative Staphylococci (CoNS)* for modes of transport as they differed significantly too. *Staphylococcus aureus* also was considerably higher in Micro Buses 22 (42.31%) than AC Buses 10 (19.23%) and Wingers 20 (38.46%) (P=0.012). *Proteus spp.* occurred much more in Wingers 6 (46.15%) than AC Buses 2 (15.38%) and Micro Buses 5 (38.46%) (P=0.023). *Pseudomonas spp.* occurred significantly more in Micro Buses 7 (53.85%) than Wingers 5 (38.46%) and AC Buses 1 (7.69%) (P<0.001).*Citrobacter spp.* occurred significantly more in Micro Buses (N=4), (44.44%) and Wingers (N=4) (44.44%) than in AC Buses (N=1) (11.11%) (p=0.026) (Table 1).

## Distribution of bacterial isolates amongst various surfaces (steering-S, handle-H, seat back-SB, seat handle-SH)

Distribution of bacterial isolates across different sampling sites, namely Steering-S (steering wheels), Handle-H (handles), Seat back-SB (seat backs), and Seat handle-SH (seat handles). Each column represents the percentage of specific bacterial species found at the corresponding sampling site. For instance, in the Steering-S category, *Coagulase-negative staphylococci (CoNS)* constitute 13 (18.06%) of isolates, while *Staphylococcus aureus (S. aureus)* accounts for 11 (21.15%). Similarly, the Handle-H site shows a prevalence of 19 (26.39%) for *CoNS* and 13 (25.00%) for S. aureus. Notably, significant variations in bacterial distribution are observed across sampling sites, as indicated by the calculated

**Table 1. Distribution of bacterial isolates between AC bus, micro bus, and winger.**

| Bacteria | AC Bus (n=27) | Microbus (n=27) | Winger (n=27) | Total (n=81) | P-value |
|---|---|---|---|---|---|
| *Citrobacter spp.* | 1 (11.11%) | 4 (44.44%) | 4 (44.44%) | 9 (11.11%) | 0.026 |
| *Acinetobacter spp.* | 2 (20.00%) | 3 (30.00%) | 5 (50.00%) | 10 (12.35%) | 0.236 |
| *Micrococcus spp.* | 3 (27.27%) | 5 (45.45%) | 3 (27.27%) | 11 (13.58%) | 0.334 |
| *Enterobacter spp.* | 3 (27.27%) | 3 (27.27%) | 5 (45.45%) | 11 (13.58%) | 0.332 |
| *Bacillus spp.* | 3 (23.08%) | 4 (30.77%) | 6 (46.15%) | 13 (16.05%) | 0.186 |
| *Pseudomonas spp.* | 1 (7.69%) | 7 (53.85%) | 5 (38.46%) | 13 (16.05%) | <0.001 |
| *Proteus spp.* | 2 (15.38%) | 5 (38.46%) | 6 (46.15%) | 13 (16.05%) | 0.023 |
| *Klebsiella spp.* | 6 (23.08%) | 12 (46.15%) | 8 (30.77%) | 26 (32.10%) | 0.088 |
| *Escherichia coli* | 8 (25.00%) | 14 (43.75%) | 10 (31.25%) | 32 (39.51%) | 0.057 |
| *Staphylococcus aureus* | 10 (19.23%) | 22 (42.31%) | 20 (38.46%) | 52 (64.20%) | 0.012 |
| *CoNS* | 14 (19.44%) | 32 (44.44%) | 26 (36.11%) | 72 (88.89%) | <0.001 |

This table compares the prevalence of bacterial isolates found in different types of public transportation.

**Abbreviations:** AC= Air Conditioner, *spp* = Species, n= Total Number.

P-values. For instance, the Seat back-SB and Seat handle-SH sites exhibit notably low P-values of 0.002 and 0.001, respectively, suggesting a statistically significant association between bacterial isolates and these particular sampling locations (Table 2).

### Antibiotics susceptibility profile for Gram-positive isolates

*Coagulase-negative Staphylococci (CoNS)* strains exhibit high sensitivity to Gentamicin 70(97.22%) and Ciprofloxacin 58(80.56%), contrasting with lower sensitivity to Piperacillin/tazobactam27 (37.50%). *Staphylococcus aureus* strains show comparable high sensitivity to Ciprofloxacin 44(84.62%) and Gentamicin 41(78.85%) yet demonstrate decreased sensitivity to Erythromycin 22(42.31%). Bacillus species display relatively high sensitivity to Gentamicin 11(84.62%) and Ceftriaxone 10(76.92%), while *Micrococcus* species exhibit exceptional sensitivity to Gentamicin 12(100.00%). Conversely, Piperacillin/tazobactam consistently reveals lower efficacy across bacterial species (Table 3).

### Antibiotics susceptibility profile for Gram-negative isolates

Amikacin shows high sensitivity percentages against *Escherichia coli* 22 (68.8%), *Klebsiella spp.* 22 (84.6%), *Pseudomonas spp.*13 (100.0%), *Proteus spp.* 8 (61.5%), *Enterobacter spp.*11 (100.0%), *Acinetobacter spp.*9 (90.0%), and *Citrobacter spp.*8 (88.9%). Meropenem exhibits similarly high sensitivity percentages across the board, ranging from 88.5% to 100.0%, indicating efficacy against all tested organisms. Gentamicin also displays excellent sensitivity percentages, with 90.6% to 100.0% sensitivity across all organisms (Table 4).

For instance, amoxicillin-clavulanic acid shows relatively high resistance percentages against several organisms, ranging from 46.2% to 69.2%. Co-trimoxazole also demonstrates notable resistance percentages, ranging from 23.1% to 44.4%, indicating a significant proportion of organisms resistant to this antibiotic. Additionally, ceftriaxone exhibits high resistance percentages against most organisms, with 61.5% to 100.0% resistance observed.

### Discussion

This study investigated the level of bacterial contamination and antibiotic resistance patterns found in public vehicles of Butwal Nepal to understand whether public transport can act as a reservoir for pathogens. It was found that 80.86% of swab samples were positive for bacterial growth and the identification stage revealed both Gram-positive and Gram-negative organisms, with *Coagulase-negative staphylococci*, and *Staphylococcus aureus* being the most prevalent

**Table 2. Distribution of Bacterial isolates amongst various surfaces.**

| Bacteria | Steering (S) | Handle (H) | Seat Back (SB) | Seat Handle (SH) | Total |
|---|---|---|---|---|---|
| *Citrobacter spp.* | 3 | 2 | 2 | 2 | 9 |
| *Acinetobacter spp.* | 2 | 2 | 2 | 4 | 10 |
| *Micrococcus spp.* | 3 | 1 | 3 | 4 | 11 |
| *Enterobacter spp.* | 1 | 3 | 3 | 4 | 11 |
| *Bacillus spp.* | 2 | 5 | 1 | 5 | 13 |
| *Pseudomonas spp.* | 3 | 4 | 1 | 5 | 13 |
| *Proteus spp.* | 4 | 4 | 4 | 1 | 13 |
| *Klebsiella spp.* | 5 | 8 | 7 | 6 | 26 |
| *Escherichia coli* | 11 | 9 | 2 | 10 | 32 |
| *Staphylococcus aureus* | 11 | 13 | 16 | 12 | 52 |
| *CoNS* | 13 | 19 | 18 | 22 | 72 |

This table details the distribution of bacterial isolates found on various surfaces: steering (S), handle (H), seat back (SB), and seat handle (SH).

**Abbreviations:** *CoNS = Coagulase-negative staphylococci, spp = Species*.

**Table 3. Antibiotics susceptibility profile for Gram-positive isolates.**

| SN | Antibiotics | CoNS (%,n=72,S/R) | S. aureus (%,n=52, S/R) | Bacillus spp (%,n=13, S/R) | Micrococcus spp (%,n=12, S/R) |
|---|---|---|---|---|---|
| 1 | Amikacin | 63.89 (46/26) | 61.54 (32/ 20) | 69.23 (9/ 4) | 66.67 (8/4) |
| 2 | Cefoxitin | 56.94 (41/31) | 63.46 (33/19) | 46.15 (6/7) | 50.00 (6/6) |
| 9 | Ceftriaxone | 69.44 (50/22) | 57.69 (36/16) | 76.92 (10/3) | 75.00 (9/ 3) |
| 3 | Chloramphenicol | 76.39 (55/17) | 67.31 (35/17) | 61.54 (8/5) | 58.33 (7/5) |
| 4 | Ciprofloxacin | 80.56 (58/14) | 84.62 (44/8) | 69.23 (9/3) | 83.33 (10/2) |
| 10 | Co-trimoxazole | 76.39 (55/17) | 73.08 (40/12) | 38.46 (5/8) | 66.67 (8/4) |
| 7 | Erythromycin | 56.94 (41/31) | 42.31 (22/30) | 23.08 (3/10) | 41.67 (7/5) |
| 8 | Gentamicin | 97.22 (70/2) | 78.85 (41/11) | 84.62 (11/2) | 100.00 (12/0) |
| 5 | Imipenem | 50.00 (36/36) | 67.31 (35/17) | 69.23 (9/4) | 75.00 (9/3) |
| 6 | Piperacillin/tazobactam | 37.50 (27/45) | 80.77 (42/10) | 76.92 (10/3) | 58.33 (5/7) |

This table presents the susceptibility patterns of Gram-positive bacterial isolates to various antibiotics.

**Abbreviations:** *CoNS = Coagulase-negative staphylococci, spp = Species*, S = Sensitive, R = Resistant, % = Percentage, n = Total Number.

**Table 4. Antibiotic susceptibility profile for Gram-negative isolates.**

| Antibiotics | *Escherichia coli* | *Klebsiella spp.* | *Pseudomonas spp.* | *Proteus spp.* | *Enterobacter spp.* | *Acinetobacter spp.* | *Citrobacter spp.* |
|---|---|---|---|---|---|---|---|
| | (%,n=32, S/R) | (%,n=26, S/R) | (%,n=13, S/R) | (%,n=13, S/R) | (%,n=11, S/R) | (%,n=10, S/R) | (%, n=9, S/R) |
| Amikacin | 68.8 (22/10) | 84.6 (22/4) | 100.0 (13/0) | 61.5 (8/5) | 100.0 (11/0) | 90.0 (9/1) | 88.9 (8/1) |
| Amoxicillin-clavulanic | 34.4 (11/21) | 46.2 (12/14) | 69.2 (9/4) | 38.5 (8/5) | 22.2 (2/7) | 20.0 (2/8) | 22.2 (2/7) |
| Cefotaxime | 43.8 (14/18) | 38.5 (10/16) | 92.3 (12/1) | 76.9 (10/3) | 100.0 (11/0) | 100.0 (10/0) | 55.6 (5/4) |
| Ceftazidime | 53.1 (17/15) | 53.8 (14/12) | 46.2 (6/7) | 69.2 (9/4) | 36.4 (4/7) | 60.0 (6/4) | 44.4 (4/5) |
| Ceftriaxone | 37.5 (12/20) | 61.5 (16/10) | 69.2 (9/4) | 76.9 (10/3) | 100.0 (11/0) | 100.0 (10/0) | 100.0 (9/0) |
| Chloramphenicol | 81.3 (26/6) | 46.2 (12/14) | 84.6 (11/2) | 100.0 (13/0) | 90.9 (10/1) | 50.0 (5/5) | 66.7 (6/4) |
| Ciprofloxacin | 90.6 (29/3) | 73.1 (19/7) | 100.0 (13/0) | 76.9 (10/3) | 100.0 (11/0) | 100.0 (10/0) | 100.0 (9/0) |
| Co-trimoxazole | 31.3 (10/22) | 38.5 (10/16) | 23.1 (3/10) | 38.5 (5/8) | 8.3 (1/12) | 30.0 (3/7) | 44.4 (4/5) |
| Gentamycin | 90.6 (29/3) | 100.0 (26/0) | 100.0 (13/0) | 100.0 (12/0) | 100.0 (11/0) | 100.0 (11/0) | 100.0 (9/0) |
| Meropenem | 90.6 (29/3) | 88.5 (23/3) | 100.0 (13/0) | 100.0 (13/0) | 100.0 (11/0) | 100.0 (11/0) | 100.0 (9/0) |

This table presents the susceptibility patterns of Gram-negative bacterial isolates to various antibiotics.

**Abbreviations:** *CoNS = Coagulase-negative staphylococc, spp = Species*, S = Sensitive, R = Resistant, % = Percentage, n = Total Number.

Gram-positive isolates and *Escherichia coli* and *Klebsiella spp.* being the most prevalent Gram-negative organisms. The distribution of bacteria differed greatly from vehicle to vehicle, and from surface to surface. It was documented that there was strong resistance to β-lactams and co-trimoxazole, but there was still efficacy of aminoglycosides and carbapenems. The study acheived its objectives that addressed major public health concerns in the urban environment of microbial contamination and antibiotic resistant organisms in a shared transporation situation.

The levels of bacterial contamination reported in our study were similar to those reported in other countries. For example, in Accra, Ghana, Darko et al. (2021) surveyed bus surfaces and found high levels of bacterial load, including *Staphylococcus spp.* and enteric bacteria [2]. Birteksoz Tan and Erdogdu (2017) also reported high microbial loads on handles of public vehicles in Istanbul, Turkey, suggesting that high touch surfaces are susceptible to high levels of contamination [3]. Similar findings were reported in the Kathmandu Valley, Nepal, where Angbuhang et al. (2018) found high levels of microbial contamination on bus surfaces, which were attributed to poor levels of enforcement on sanitary practices [8]. These studies, taken as a whole, confirm that bacterial contamination of public transport vehicles is a global issue, that hygiene and sanitation policies are highly related to control of bacterial contamination.

Our study identified *S. aureus* and *coagulase-negative Staphylococci (CoNS)* as the most common Gram-positive isolates. This is in line with a Ghanaian study where *S. aureus*, including MRSA, was frequently isolated from public transport, raising concerns about infection risks [5]. Similarly, Kahsay et al. (2019) in Ethiopia reported both enteric bacteria and MRSA from bus surfaces, suggesting that public transport environments harbor a mixture of commensals and pathogenic organisms [12]. These results collectively highlight the role of public vehicles as reservoirs for community- and healthcare-associated pathogens, with the potential to facilitate their widespread transmission.

Among Gram-negative bacteria, *E. coli* and *Klebsiella spp.* were most frequently isolated in our study, pointing to fecal contamination and inadequate sanitation. Comparable findings were reported by UjJaman Arowan et al. (2021) in Dhaka, Bangladesh, where *E. coli* and other enteric pathogens were frequently detected on public transport [11]. Moreover, Fernanda et al. (2023) detected β-lactam resistance genes in bacterial isolates from Quito, Ecuador, demonstrating that transport systems may also serve as reservoirs of resistance determinants [4]. A similar study in Seoul, South Korea revealed that microbial communities on public transit carried multidrug resistance genes (Jang et al., 2022) [15]. Taken together, these findings emphasize that public transport not only facilitates bacterial transmission but also may contribute to the spread of antimicrobial resistance (AMR).

Our surface-specific findings had significant correlations with certain surfaces, particularly high-touch surfaces, including seatbacks and seat handles. Previous research from Europe has highlighted door handles and railings as hotspots for microbial risk, and highlighted concerns for these surfaces, resulting from microbial contamination from passengers [7,18]. Our data highlights the critical need for cleaning protocols for all transport vehicle surfaces, particularly high-touch surfaces, to minimize exposure risk.

In our study, antibiotic susceptibility testing indicated finding that both aminoglycosides (amikacin and gentamicin) and carbapenems (meropenem) were still the most effective agents against both Gram-positive and Gram-negative isolates. It has been reported that in Austria and India despite the well-documented problem of widespread multidrug resistance there was still a pattern of maintained sensitivity to aminoglycosides and carbapenems [16,19]. However, the high rates of resistance to β-lactams and co-trimoxazole that we found in our isolates correspond to the findings out of Ghana and Vienna that reported *Enterobacteriaceae* exhibiting resistance rates in excess of 50% [6,19]. Taken together these findings emphasize the growing concern of AMR in public spaces, likely a result of inappropriate antibiotic use, and insufficient environmental sanitation [13,28].

Overall the results of this study have meaningful implications for public health. Public transport vehicles serve as a vehicle and vector for transmissible pathogens and antibiotic resistant organisms; the fact that public transport represents an opportunity for common community spread is a significant findings together with integrative reviews that propose public transport to be a neglected but notable contributing factor to community-acquired infections [9,27]. This is particularly

important for Nepal where public transport is deeply relied upon, and the results would urgently necessitate interventions such as disinfecting frequently used trappable surfaces on scheduled basis, education campaigns for the community, being aware of potential hygiene standards based exposure parameters in industry best practices and regulations for transport vehicles [20,29]. Additionally, ongoing reporting of resistance and ongoing surveillance in the environment for understanding and consideration in addition to clinical surveillance programs will be necessary [20,28]. This study provides useful insights into public health planning and development of targeted interventions to reduce microbial hazards and improve hygiene behaviours.

## Limitations

There are a few limitations that must be acknowledged when evaluating the results of this study. To begin, the study was conducted in a single city (Butwal), and thus the results may not reflect microbial contamination patterns across all areas of Nepal. Second, while we discovered a variety of bacteria, we did not employ molecular methods (for example, PCR and sequencing) which would have provided a more precise identification and characterization for resistance genes. Third, the cross-sectional approach limits ability to investigate temporal changes in bacterial contamination across geographically distinct areas, and with different seasons. Fourth, the study focused only on bacterial contaminants and did not investigate the presence of other possible viral or fungal pathogens or contributants to transport related infections. In light of the limitations noted above, this study provides an essential baseline from which to begin to understand the microbial risks presented by public transport systems in Nepal.

## Recommendations

To reduce the risk of bacterial contamination and antimicrobial resistance from public transport systems, immediately after a commitment to routine disinfecting of high touch surfaces should made, and national hygiene standards are agreed upon, monitoring of compliance with the agreed international standards is recommended. Public campaigns to improve awareness can encourage compliance with preventive behaviours, including washing hands and using hand sanitizers while travelling. Increasing public health protection can also be supported by introducing antimicrobial resistance surveillance into current National antimicrobial resistance action plans, as well as aspects of transport hygiene being included in infection prevention and stewardship plans. An attention to capacity building activities such as training and incentivising transport workers can be undertaken to contribute to meaningful sanitation improvements.

## Conclusion

This study indicates that public transportation in Butwal, Nepal, presents a significant public health risk. Bacterial contamination levels were found to be high in publicly available transport vehicles surfaces, and that these surfaces had both Gram-positive and Gram-negative pathogens. The presence of multi-drug resistant bacteria also indicated the potential for public transport vehicles to act as a reservoir and vehicle for the transmission or spread of antimicrobial resistance into the community. In general, the results from this study indicated that public health interventions are needed to improve sanitation and disinfection of public transport, uphold hygiene behaviours related to transport, and implement AMR surveillance as part of public health practice. It will be important to focus attention on public health interventions in Butwal and other cities, through the use of evidence based, continuing community awareness, in order to address these issues to create a safer environment to diminish risk of infection for public health in Nepal. This issue will also require collective action from agencies of government, owners and operators of public transportation, and the community to ensure that hygiene standards are maintained, and the impacts of antimicrobial resistance are limited in their community.

## Supporting information

**S1 Data. Data analysis.** This dataset summarizes the analytical methods employed, including descriptive statistics, chi-square tests, and the statistical significance of relationships between variables such as public vehicle type and antibiotic resistance patterns.
(XLSX)

## Acknowledgments

The author would like to acknowledge the Pokhara University Research Center (PURC) and Crimson College of Technology (CCT) for their kind support and guidance during research work.

## Author contributions

**Conceptualization:** Ram Bahadur Khadka, Balram Neupane.

**Data curation:** Khimdhoj Karki.

**Funding acquisition:** Ram Bahadur Khadka.

**Investigation:** Saksham Pokharel.

**Methodology:** Khimdhoj Karki.

**Project administration:** Ram Bahadur Khadka.

**Resources:** Balram Neupane.

**Software:** Ram Bahadur Khadka.

**Supervision:** Balram Neupane.

**Validation:** Ram Bahadur Khadka, Saksham Pokharel.

**Visualization:** Balram Neupane.

**Writing – original draft:** Saksham Pokharel.

**Writing – review & editing:** Khimdhoj Karki.

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
