## [Decision Letter · Decision Letter 0]

12 Aug 2025

PGPH-D-25-01435

Bacterial Diversity and Antibiotic Susceptibility Profile of the Isolates from Public Vehicles in Butwal Sub-Metropolitan City, Rupandehi, Nepal

Dear Dr. KHADKA,

Thank you for submitting your manuscript to PLOS Global Public Health. After careful consideration, we feel that it has merit but does not fully meet PLOS Global Public Health’s publication criteria as it currently stands. Therefore, we invite you to submit a revised version of the manuscript that addresses the points raised during the review process.

Please review in detail each comment from both reviewers, both those contained in the attached file and those included in the text of this email, and provide an explicit, point-by-point response indicating how each comment is addressed in the revised manuscript.

We look forward to receiving your revised manuscript.

Kind regards,

Delfina Fernandes Hlashwayo, Ph.D.

Academic Editor

Journal Requirements:

1. We have amended your Competing Interest statement to comply with journal style. We kindly ask that you double check the statement and let us know if anything is incorrect.

2. Please provide a/amend your detailed Financial Disclosure statement. This is published with the article. It must therefore be completed in full sentences and contain the exact wording you wish to be published.

**Please only choose the relevant sentences from below**

a. Please clarify all sources of funding (financial or material support) for your study. List the grants (with grant number) or organizations (with url) that supported your study, including funding received from your institution.

b. State the initials, alongside each funding source, of each author to receive each grant.

c. State what role the funders took in the study. If the funders had no role in your study, please state: “The funders had no role in study design, data collection and analysis, decision to publish, or preparation of the manuscript.”

d. If any authors received a salary from any of your funders, please state which authors and which funders.

3. In the online submission form, you indicated that “The data supporting the findings of this study are available upon request from the corresponding author”.

a. In a public repository,

b. Within the manuscript itself, or

c. Uploaded as supplementary information.

Additional Editor Comments (if provided):

Reviewers' comments:

Reviewer's Responses to Questions

**Comments to the Author**

1. Does this manuscript meet PLOS Global Public Health’s publication criteria? Is the manuscript technically sound, and do the data support the conclusions? The manuscript must describe methodologically and ethically rigorous research with conclusions that are appropriately drawn based on the data presented.

Reviewer #1: Partly

Reviewer #2: Yes

2. Has the statistical analysis been performed appropriately and rigorously?

Reviewer #1: Yes

Reviewer #2: Yes

3. Have the authors made all data underlying the findings in their manuscript fully available (please refer to the Data Availability Statement at the start of the manuscript PDF file)?

Reviewer #1: No

Reviewer #2: No

4. Is the manuscript presented in an intelligible fashion and written in standard English?

Reviewer #1: Yes

Reviewer #2: Yes

5. Review Comments to the Author

Reviewer #1: During the reading of the manuscript, I noted several positive aspects, including the clarity of presentation, the use of accessible language by the authors, and the overall logical structure of the manuscript, which aligns well with the requirements for publication in PLOS Global Public Health.

However, there are several areas that require improvement or clarification. I highlight the following:

• Methodology (Line 97): The authors mention the tests used for bacterial identification and refer to the Bergey’s Manual of Bacteriology for further procedural details. However, they do not describe how these tests were conducted in the context of their study. It would be more appropriate to briefly describe the procedures rather than simply citing the manual.

• Line 112: Although the antimicrobial susceptibility testing (AST) procedure is generally well described, it would be beneficial to include more specific details about the incubation conditions, particularly the temperature and duration.

• Line 115: The authors state that the diameters of inhibition zones around each disc were measured and interpreted according to CLSI guidelines. However, it would be helpful to specify the exact criteria or breakpoints from the CLSI used for interpretation.

• Line 124 (Data analysis): The manuscript mentions the use of SPSS for data analysis, but it is unclear what specific information or variables were analyzed using this software, and how the variables were classified. Additionally, the purpose of the chi-square test should be clarified, what associations were being tested?

• Line 139 (Sampling strategy): The authors state that swab samples were collected from different parts of the buses. It would strengthen the manuscript to explain the rationale for selecting these particular sites. While readers with microbiological background may infer the reasoning (i.e., higher likelihood of contamination), this should be explicitly stated for the benefit of a broader audience.

• Line 142 (Graph of gram-negative bacteria): It is suggested that the bacterial frequency values be reordered from lowest to highest (or vice versa), for consistency and improved readability, as was done with gram-positive bacteria.

• Graph design: In the graph showing the distribution of bacteria, it would be clearer to use distinct colors to differentiate between sampling locations such as the steering wheel and seat handle.

• Results section: While the manuscript describes the bacterial analysis, no images or visual documentation of the findings are provided. Including representative images (e.g., culture plates or microscopic images) would add value and help validate the results.

• Ethics approval: Although the authors state that ethical approval was obtained from the Pokhara University Research Center, the ethics approval number is missing. As per the journal’s policy, this information should be clearly included in the manuscript.

• References: The journal follows the Vancouver referencing style, but several references in the manuscript require formatting corrections, particularly regarding journal titles, article titles, punctuation, and consistency in DOI or URL presentation. Adjustments are necessary to ensure full compliance with the journal’s style guide.

Reviewer #2: I recommend revision of the manuscript. Due to the detailed nature of my comments, I have included them in the attached file. The comments address issues related to citation accuracy, clarity of the results and discussion section, reference formatting, and data presentation. I encourage the authors to review the suggestions carefully and revise the manuscript accordingly

6. PLOS authors have the option to publish the peer review history of their article (what does this mean?). If published, this will include your full peer review and any attached files.

**Do you want your identity to be public for this peer review?** For information about this choice, including consent withdrawal, please see our Privacy Policy.

Reviewer #1: **Yes:**Nicole De Sérgio João Olim Freitas

Reviewer #2: No

---

## [Decision Letter · Decision Letter 1]

20 Oct 2025

PGPH-D-25-01435R1

Bacterial Diversity and Antibiotic Susceptibility Profile of the Isolates from Public Vehicles in Butwal Sub-Metropolitan City, Rupandehi, Nepal

Dear Dr. KHADKA,

Thank you for submitting your manuscript to PLOS Global Public Health. After careful consideration, we feel that it has merit but does not fully meet PLOS Global Public Health’s publication criteria as it currently stands. Therefore, we invite you to submit a revised version of the manuscript that addresses the points raised during the review process.

We look forward to receiving your revised manuscript.

Kind regards,

Delfina Fernandes Hlashwayo, Ph.D.

Academic Editor

Journal Requirements:

Additional Editor Comments (if provided):

Reviewers' comments:

Reviewer's Responses to Questions

**Comments to the Author**

1. If the authors have adequately addressed your comments raised in a previous round of review and you feel that this manuscript is now acceptable for publication, you may indicate that here to bypass the “Comments to the Author” section, enter your conflict of interest statement in the “Confidential to Editor” section, and submit your "Accept" recommendation.

Reviewer #1: All comments have been addressed

Reviewer #2: (No Response)

2. Does this manuscript meet PLOS Global Public Health’s publication criteria? Is the manuscript technically sound, and do the data support the conclusions? The manuscript must describe methodologically and ethically rigorous research with conclusions that are appropriately drawn based on the data presented.

Reviewer #1: Partly

Reviewer #2: Partly

3. Has the statistical analysis been performed appropriately and rigorously?

Reviewer #1: Yes

Reviewer #2: Yes

4. Have the authors made all data underlying the findings in their manuscript fully available (please refer to the Data Availability Statement at the start of the manuscript PDF file)?

Reviewer #1: No

Reviewer #2: Yes

5. Is the manuscript presented in an intelligible fashion and written in standard English?

Reviewer #1: Yes

Reviewer #2: Yes

6. Review Comments to the Author

Reviewer #1: The manuscript presents improvements in methodology, statistical analysis, clarity, ethics, and overall formatting, reflecting significant effort by the authors. However, there are still minor gaps that must be addressed for publication in PLOS Global Public Health.

Methodology (point 97): The authors added details on the tests used for bacterial identification, according to Bergey's Manual of Bacteriology, which shows that the methodology is now well described and follows expected standards.

In point 112: In the AST procedure for isolated bacteria, the text now specifies that the plates were incubated at 35 ± 2°C for 16–18 hours, which was previously not mentioned in this point.

In point 115: The specific criteria used to interpret the diameter of the inhibition zones around each disk and how this was measured were described, specifying the susceptible, intermediate, and resistant categories, according to CLSI guidelines.

In point 124: Which corresponds to data analysis, the authors explain that the data were processed in SPSS 21.0, using descriptive statistics and the chi-square test to assess associations between categorical variables (e.g., vehicle type and bacterial presence), which was not previously mentioned.

In point 139: Which is the results section, the authors mention having collected samples from different contact surfaces; however, they continue without justifying the reason for choosing these locations for the lay reader. I recommend including a brief sentence explaining that these locations were chosen because they are the surfaces with the highest risk of contamination due to frequent contact with passengers.

In point 142: I suggested that, in the frequency graph of gram-negative bacteria, their total values follow an order from lowest to highest, and so on, for better organization, as in the gram-positive bacteria section. However, the graph remains without the ordering of the values as suggested.

Graphic design: In graph 2, which shows the distribution of bacteria on different surfaces, I suggested color-coding the steering wheel and seat handle, which would facilitate visual identification. Therefore, I recommend this change.

Results section: In the bacterial analysis, no images are provided to support or represent the results. Including these would be important to validate the results presented. According to the journal's guidelines, images should be uploaded in a separate file from the manuscript, allowing them to be included as supplementary material.

Ethics approval: The authors added details regarding the ethics approval number (02-2079/80), along with a note that no human or animal data were used.

References: The references have been revised and largely follow the Vancouver format, but there are still minor duplications and inconsistencies in some links.

Reviewer #2: After a detailed review of the revised manuscript, I found that the comments and suggestions provided by the reviewers have not been adequately addressed. Furthermore, a response document to the reviewers was not submitted. The manuscript does not show significant changes in relation to the critical observations made, and, for this reason, it cannot be recommended for publication in its current version.

7. PLOS authors have the option to publish the peer review history of their article (what does this mean?). If published, this will include your full peer review and any attached files.

**Do you want your identity to be public for this peer review?** For information about this choice, including consent withdrawal, please see our Privacy Policy.

Reviewer #1: No

Reviewer #2: No

Figure Resubmissions:

---

## [Decision Letter · Decision Letter 2]

25 Nov 2025

PGPH-D-25-01435R2

Bacterial Diversity and Antibiotic Susceptibility Profile of the Isolates from Public Vehicles in Butwal Sub-Metropolitan City, Rupandehi, Nepal

Dear Dr. KHADKA,

Thank you for submitting your manuscript to PLOS Global Public Health. After careful consideration, we feel that it has merit but does not fully meet PLOS Global Public Health’s publication criteria as it currently stands. Therefore, we invite you to submit a revised version of the manuscript that addresses the points raised during the review process.

Please address the reviewer's concerns in your revised manuscript, providing a point-by-point response to their comments upon resubmission.

We look forward to receiving your revised manuscript.

Kind regards,

Sarah Jose, Ph.D.

Staff Editor

Journal Requirements:

Additional Editor Comments (if provided):

Reviewers' comments:

Reviewer's Responses to Questions

**Comments to the Author**

1. If the authors have adequately addressed your comments raised in a previous round of review and you feel that this manuscript is now acceptable for publication, you may indicate that here to bypass the “Comments to the Author” section, enter your conflict of interest statement in the “Confidential to Editor” section, and submit your "Accept" recommendation.

Reviewer #2: (No Response)

2. Does this manuscript meet PLOS Global Public Health’s publication criteria? Is the manuscript technically sound, and do the data support the conclusions? The manuscript must describe methodologically and ethically rigorous research with conclusions that are appropriately drawn based on the data presented.

Reviewer #2: Partly

3. Has the statistical analysis been performed appropriately and rigorously?

Reviewer #2: Yes

4. Have the authors made all data underlying the findings in their manuscript fully available (please refer to the Data Availability Statement at the start of the manuscript PDF file)?

Reviewer #2: No

5. Is the manuscript presented in an intelligible fashion and written in standard English?

Reviewer #2: (No Response)

6. Review Comments to the Author

Reviewer #2: Thank you for the current version. the manuscript has improved significantly. In the previous version, the manuscript had continuous line numbering, but it was not applied throughout the entire document, in the current version, there is no line numbering at all, which makes the revision process difficult. I kindly request that the authors apply continuos line numbering to the entire manuscript.

Please specify exactly in which repository the database is available, and, if possible, provide the acess link to the database.

Review the materials an methods section and clarify whether the study period was from June 2023 to July 2024, or from June 2021 to February 2022 as stated in the response letter. There is discrepancy between the two description.

Review the reference list, as it requires more attetion and clarify.

7. PLOS authors have the option to publish the peer review history of their article (what does this mean?). If published, this will include your full peer review and any attached files.

**Do you want your identity to be public for this peer review?** For information about this choice, including consent withdrawal, please see our Privacy Policy.

Reviewer #2: No

Figure Resubmissions:

---

## [Editor Report · Decision Letter 3]

26 Dec 2025

PGPH-D-25-01435R3Bacterial Diversity and Antibiotic Susceptibility Profile of the Isolates from Public Vehicles in Butwal Sub-Metropolitan City, Rupandehi, NepalPLOS Global Public Health

Dear Dr. KHADKA,

Thank you for submitting your manuscript to PLOS Global Public Health. After careful consideration, we have decided that your manuscript does not meet our criteria for publication and must therefore be rejected.

We are sorry that we cannot be more positive on this occasion. We very much appreciate your wish to present your work in one of PLOS's Open Access publications. Thank you for your support, and we hope that you will consider PLOS Global Public Health for other submissions in the future.

Yours sincerely,

Jay-ar Formentera Medes

Support Staff - Editorial
---

## [Editor Report · Decision Letter 4]

20 Apr 2026

Bacterial Diversity and Antibiotic Susceptibility Profile of the Isolates from Public Vehicles in Butwal Sub-Metropolitan City, Rupandehi, Nepal

PGPH-D-25-01435R4

Dear Mr KHADKA,

We are pleased to inform you that your manuscript 'Bacterial Diversity and Antibiotic Susceptibility Profile of the Isolates from Public Vehicles in Butwal Sub-Metropolitan City, Rupandehi, Nepal' has been provisionally accepted for publication in PLOS Global Public Health.

Best regards,

Julia Robinson

Executive Editor